# Hospital Dentistry for Intensive Care Unit Patients: A Comprehensive Review

**DOI:** 10.3390/jcm10163681

**Published:** 2021-08-19

**Authors:** Mi-Kyoung Jun, Jeong-Kui Ku, Il-hyung Kim, Sang-Yoon Park, Jinson Hong, Jae-Young Kim, Jeong-Keun Lee

**Affiliations:** 1Department of Oral and Maxillofacial Surgery, Institute of Oral Health Science, Ajou University School of Medicine, Suwon 16499, Korea; mijjomg@naver.com; 2Department of Oral and Maxillofacial Surgery, Gangnam Severance Hospital, Yonsei University College of Dentistry, Seoul 06273, Korea; KUJK@yuhs.ac (J.-K.K.); KJY810927@yuhs.ac (J.-Y.K.); 3Department of Oral and Maxillofacial Surgery, Armed Forces Capital Dental Hospital, Armed Forces Medical Command, Seongnam 13574, Korea; haonflower@gmail.com (I.-h.K.); psypjy0112@naver.com (S.-Y.P.); 4Department of Prosthodontics, Armed Forces Capital Dental Hospital, Armed Forces Medical Command, Seongnam 13574, Korea; kma55mir@gmail.com

**Keywords:** hospital dentistry, intensive care unit, oral care, oral health, oral hygiene

## Abstract

This study aimed to review the oral hygiene status, oral care guidelines, and outcomes of oral care in intensive care unit (ICU) patients from a dental perspective for effective oral care. A literature search using the keywords “Hospital dentistry” OR “Oral care” OR “Intensive care unit” OR “Hospital inpatient” OR “Hospitalization” OR “Emergency service” AND “Oral health” OR “Oral hygiene” OR “Dental plaque” was conducted in PubMed, Medline, and Google Scholar to identify publications reporting on the oral care of the patients admitted to ICUs. A total of 17,400 articles were initially identified. Of these, 58 were selected and classified into three categories for critical review. Seven of these studies evaluated the oral status of ICU patients, and most of the studies indicated that ICU patients had poor oral hygiene or required active dental treatment. Thirty-three of these studies evaluated oral care methods for ICU patients, and in general, oral care methods using chlorhexidine as adjuncts along with tooth brushing were recommended. However, there were insufficient studies to evaluate oral hygiene through effective assessment tools from a dental perspective. In 36 studies on the outcomes of oral care in ICU patients, interventions by dental professionals showed effective results in preventing hospital-acquired infection. This review highlights the importance of establishing guidelines for the evaluation of oral status in ICU patients and summarizes data that may be useful for future studies. Further studies on maintaining good oral hygiene among ICU patients are needed.

## 1. Introduction

Insufficient oral hygiene promotes plaque accumulation and colonization by pathogenic bacteria, which facilitate the dissemination of pathogens [1]. Poor oral hygiene is known to increase the risk of pathology in other organs, such as the respiratory system [2,3]. Several studies have revealed that the oral hygiene status of intensive care unit (ICU) patients affects the occurrence of ventilator-associated pneumonia (VAP) [4,5,6]. VAP is the most common cause of hospital-acquired infection (HAI) in the ICU setting and is the second most common nosocomial infection. It is a serious medical condition with a risk of mortality of 33–50% and is highly associated with intraoral bacteria that colonize dental plaque and calculus [7].

In 2018, de Carvalho Baptista et al. showed that the presence of microorganisms in the oral cavity, including *Enterococcus faecalis*, *Fusobacterium periodonticum*, *Gemella morbillorum*, *Neisseria mucosa*, *Propionibacterium acnes*, *Prevotella melaninogenica*, *Streptococcus oralis*, *Streptococcus sanguinis*, *Treponema denticola*, *Treponema socransckii*, and *Veillonella parvula*, were related to the increased amounts of bacteria obtained from the respiratory tracts of patients with long-term use of mechanical ventilation, and intubation could act as a pathway for migration of the oral flora to the lungs, contributing to the occurrence of VAP [8,9]. Several studies have revealed that active oral care interventions improve oral hygiene status and may reduce the risk of VAP [6]. In addition, a systematic review and meta-analysis demonstrated that the risk of non-VAP (community-acquired pneumonia—CAP, healthcare-associated pneumonia—HCAP, and hospital-acquired pneumonia—HAP) could be reduced by professional dental treatment [10]. Therefore, comprehensive oral care is essential for HAI prevention. The most common method for oral care in ICU patients is the removal of the bacteria-rich oropharyngeal secretions for preventing an aspiration into the lungs. Various oral care methods, such as mouthwash using chlorhexidine (CHX) of a 0.12–2% concentration or 10% povidone-iodine, manual or electric toothbrushing, and mechanical cleaning, are used [11]. However, the gold standard of oral care for ICU patients has not been established yet.

ICU patients usually suffer from dry mouth and oral lesions because of their medications, mastication disorders, swallowing discomfort, and difficulty in managing oral hygiene on their own. However, in previous survey studies targeting ICU nurses, 53–58% of ICU nurses answered that patients had difficulties performing oral hygiene management due to not receiving proper training, or materials and instruments not being available, and most of them responded that their oral care was neglected compared to care of other parts of the body [12,13,14,15]. According to Araujo et al., in a survey of ICU nurses, 86% of ICU nurses felt that patients needed dental treatment and 98% stated that there should be a dental team in the ICU for the oral management of patients. The nursing team suggested that oral hygiene management in the ICU is insufficient and inappropriate [16].

In 2016, Amaral et al. reported that 58.7% of ICU patients had one or more needs for invasive dental treatment including periodontal treatment, restorative treatment, surgical treatment, and endodontic treatment [17]. Another study in 2018 showed that 82.5% of ICU patients in the study required invasive treatment, and 34% required active dental treatment (incision and drainage, toothache) Furthermore, more than 62.2% of patients had periodontal disease, to the extent that oral complications occurred during hospitalization [18].

Currently, most of the evaluation criteria for the oral condition of ICU patients traditionally follow the Oral Assessment Guide proposed by Eilers [19]. This guideline was based on the subjective observations by nurses, but excluded the evaluation of periodontal diseases and dental caries, the most common oral bacterial diseases. Therefore, it is necessary to validate the Oral Assessment Guide and oral care methods used according to this guideline from the perspective of dentistry.

The aim of this review is to summarize the oral health status, oral management guidelines, and effects of oral care in ICU patients through the assessment of relevant literature published in the last decade.

## 2. Methods

An electronic search was performed in Google Scholar, Medline, and PubMed for relevant articles published between March 2011 and March 2021. The search was performed for the following keywords: “Hospital dentistry” OR “Oral care” OR “Intensive care unit” OR “Hospital inpatient” OR “Hospitalization” OR “Emergency service” AND “Oral health” OR “Oral hygiene” OR “Dental plaque.” When conducting the search, theses, textbooks, and case reports that were not written in English were excluded.

Three researchers (J.-K.K., I.-h.K. and S.-Y.P.) independently selected the studies based on the following eligibility criteria: type of intervention, type of patients, and type of study. In the first step, titles and abstracts were read for screening of articles. The full text of the selected articles was then analyzed to determine the hospital dentistry’s eligibility for the ICU. Any difference of opinions was resolved through discussion with other researchers (M.-K.J. and J.-K.L.).

Finally, the selected papers were classified into the following three types: (1) the oral status of ICU patients, (2) the comparison of oral management methods, and (3) the outcomes of oral management (Figure 1).

## 3. Results

### 3.1. Assessment of Oral Status in Intensive Care Unit Patients

A total of seven papers described the oral conditions before intervention in ICU patients [5,20,21,22,23,24]. Of these, four evaluated oral conditions from a dental perspective [5,20,21,22] (Table 1), and the rest involved evaluation using the Oral Assessment Guide proposed by Beck in 1979 and Eilers in 1988, or the modified Oral Assessment Guide proposed by Barnason et al. in 1998 [19,25,26].

In 2011, Terezakis et al. investigated the oral health of hospitalized patients through a systematic literature review and meta-analysis based on papers published between 1950 and 2010 [20]. Through a review of five randomized controlled trials (RCTs), they analyzed oral health after hospitalization by evaluating plaque accumulation, gingival inflammation, periodontal disease, and caries incidence. Three papers reported that the plaque index increased from 23% at hospitalization to 93% after 10 days of hospitalization [27], and two reported that the gingival inflammation index was significantly increased [28,29].

In an RCT to determine the effect of CHX on reducing the risk of VAP in ICU patients in 2012, residual teeth, plaque index, probing depth, and bleeding on probing (BOP) were assessed by nurses. In the study, patients lost an average of 14.3 teeth in the ICU [21]. Another study in ICU patients reported a plaque index of 85.6% and an average periodontal pocket depth of 3.8 mm, which is more than the diagnostic criterion for gingivitis (3 mm), and 48.6% of the patients had BOP, which required active periodontal treatment [34].

In 2017, Steel reviewed the studies published from 2000 to June 2017 on the oral condition of patients admitted to hospital ICUs [22]. In a study of 150 acute care patients (ages 58–94 years) in the UK, 75% of patients with natural dentition required dental operative intervention, 38% of patients wearing dentures had denture-related candidiasis, and 85% of patients had never visited a dental clinic for more than a year [30]. In a study conducted in Israel in 225 patients over 65 years of age, 65% needed direct dental treatment and 56% had pseudomembranous candidiasis [31]. In a study of 200 patients in New Zealand, 90% were in need of periodontal treatment, such as scaling, and 71% needed extraction or caries treatment and had an average of 1.9 carious teeth [32]. In a study of 575 patients within one week of hospitalization in Australia, 76% of the patients answered that their oral condition was unhealthy, and 38% indicated that they had poor oral hygiene. In addition, it was reported that the worse the oral condition, the more it was associated with dementia or moderate renal impairment [33].

In a 2018 RCT, Bellissimo-Rodrigues et al. reported that most respiratory tract infections were prevented by dentists providing active oral care (tooth brushing, tongue cleaning, scaling, root planning, caries restoration, and extraction) to ICU patients [5]. In addition, 254 patients admitted to the ICU were evaluated for oral health, and the following oral diseases were reported with high frequency: gingivitis (54.8%), completely edentulous state (38.2%), periodontitis (29.5%), dental caries (29.1%), tooth fractures requiring extraction (residual roots, 17.0%), mucositis (6.3%), oral candidiasis (1.6%), and dental abscess (0.8%). All of these diseases, except for completely edentulous state, were infectious and inflammatory conditions that could not be resolved without active dental treatment. Therefore, compared to the oral care performed by nurses using CHX, active treatment by dentists prevented more respiratory-related infections in ICU patients; therefore, active involvement of dentists is not only logically valid, but also has important clinical implications.

The modified Oral Assessment Guide was mainly used as an oral assessment tool by nurses. This guideline uses subjective observations to evaluate six areas, including the lips, mucosa, tongue, gingiva, and saliva, on a 3-point scale. The total score ranges from 6 to 18 points, with lower scores indicating better oral condition [26]. However, it is unclear how these guidelines correlate with general oral conditions, such as plaque or calculus, the depth of the periodontal pocket, BOP, and the number of remaining teeth. In a 2014 study comparing the effects of oral administration of 5% sodium bicarbonate, 0.2% CXH, and normal saline [24], the average Oral Health Assessment scores before and 4 days after intervention were 7.68 and 9.80, respectively. It was noted that all scores were between 6 and 10 points, which implied “Mild dysfunction” [19]. Subsequently, the same authors noted that there was no difference between the three types of oral function [24]. However, it was not possible to define what condition the “Mild dysfunction” state indicated in this study from a dental perspective. Therefore, there was insufficient evidence to discuss the effect of oral care from a dental perspective according to the types of oral functions as claimed.

In a 2016 RCT of ICU patients in Korea, pre-intervention conditions were analyzed using the modified Oral Assessment Guide. On average, they showed very poor oral hygiene scores for the lip (2.8), mucosa (2.5), gingiva (2.3), and saliva (2.8) [23]. The authors noted that, during oral care interventions, care should be taken to avoid gingival bleeding. Considering that gingival bleeding during hospital stay is BOP, which is a diagnostic indicator for gingivitis [34], patients with poor oral hygiene were likely to need active dental treatment for gingivitis or periodontitis because the bleeding simply cannot be resolved by brushing or applying CHX. In addition, a 0.1% CHX swab was applied with mechanical protocols, suggesting improved salivary acidity, oral moisture on mucosa, and modified Oral Assessment Guide scores, but no data were available for discussion from a dental perspective.

### 3.2. Oral Hygiene Management Methods for Patients in ICUs

The Center for Disease Control and Prevention recommends oral hygiene management as a method for preventing VAP [35]. In ICU patients on mechanical ventilation, especially those with poor oral hygiene, risk of VAP is facilitated by the accumulation of plaque. Some previous studies reported that the hospital ICU mortality rate associated with VAP was no different between the application of CHX and a toothbrush [36,37]. However, a recent review article on RCTs from 2008 to 2018 reported that VAP prevention and oral management of patients are closely related and are important factors in reducing the mortality rate of ICU patients [11]. In addition, to lower the incidence of VAP, proper oral care should be considered as a part of the medical treatment plan when patients are admitted to an ICU, and the need for oral care treatment protocols was suggested [11]. Many researchers have analyzed the effect of oral hygiene management by comparing various types of oral care products and oral care methods [38,39]. To accurately analyze this effect, an effective indicator that can objectively evaluate oral conditions is important. However, studies using the modified Oral Assessment Guide have shown that there are no differences in oral hygiene findings with different oral care methods [23,24,40]. In contrast, studies in which dental evaluation, such as plaque score, or objective evaluation, such as bacterial identification, were conducted have shown clear differences in results between various oral care methods, allowing for comparisons of the effects of oral care methods [41,42,43,44,45,46,47,48,49]. Therefore, studies must be conducted using objective evaluation indicators to identify the most effective oral management method.

In 2014, Par et al. conducted a review of the literature, listed the most commonly performed oral care methods in ICUs, and showed the advantages and disadvantages of each method (Table 2) [50]. However, there is no evidence to provide a standard guideline on the most efficient method, and although there are several evidence-based protocols, oral care guidelines are still inconsistent and vary significantly across institutions.

In general, hydrogen peroxide is irritable, has an unpleasant taste, and is genotoxic. Sodium carbonate is not recommended for use because it can cause irritation and chemical burns owing to its high pH. Tap water can be contaminated in hospitals; therefore, the use of sterile water is recommended. Topical antibiotics are not recommended because they do not have antibacterial effects against all bacteria, resulting in changes in the oral flora, and there is a risk of developing resistance. Citric acid and glycerin may temporarily relieve dry mouth, but they are not recommended because their low pH can cause hard tissue demineralization. Povidone-iodine is not effective in reducing plaque and is not recommended because it is toxic. In contrast, CHX is effective for VAP prevention and has plaque control effects at concentrations between 0.12% and 0.2%. Artificial saliva and Vaseline are recommended because they aid in moisturizing the mucosa and maintaining the physiological oral flora.

In conclusion, extensive mechanical plaque control is the most basic and efficient method for maintaining oral hygiene. It is better to use a toothbrush rather than a cotton swab for teeth, and the use of toothpaste is effective [50]. Several previous studies have suggested that brushing with a manual toothbrush and rinsing with CHX are the most commonly recommended measures for oral care in ICU patients [50,51]. However, it is often reported that uniform oral hygiene management guidelines are not implemented, even in the same institution. Goss et al. pointed out that patients on mechanical ventilation had a higher average frequency of receiving oral care than did those with spontaneous respiration, but the difference in frequency of performance was 1 to 8 h, indicating the absence of standardized oral care guidelines [52]. Efforts are being made to reduce the occurrence of VAP by providing nurses with continuous oral hygiene management education, such as on oral hygiene simulation. However, well-established theories and guidelines have not yet been prepared [3,40].

#### 3.2.1. Application of CHX

Several systematic reviews and meta-analyses have been conducted on the appropriate concentration and usage of CHX. Rabello et al. demonstrated that various CHX regimens and doses (0.12–2%) were effective in preventing nosocomial pneumonia and VAP in ICU patients [51]. In 2014, Zhang et al. compared the effects of various concentrations of CHX and concluded that 0.12% CHX was the best option with respect to cost, adverse reactions, drug resistance, and VAP prevention [53]. However, in 2019, Jackson and Owens pointed out that, although CHX shows a clear effect in preventing VAP in ICU patients, standard guidelines, such as the usage and frequency of CHX use, have not yet been established [54]. CHX demonstrates an almost 75–90% antibacterial effect even if it is applied in the oral cavity for 15 s [55]. CHX is relatively easy to apply for oral hygiene management by nurses in intensive care settings and is a mouth rinse with good antibacterial effects on pathogens in the mouth, while also effectively preventing VAP [56].

However, CHX may have side effects, such as exogenous staining, taste changes, antimicrobial resistance, burning sensation, and rarely, severe anaphylaxis [57]. Major oral diseases, periodontitis, and dental caries cannot be treated with CHX alone [58], and should be used as an effective supplement to help control plaque and gingivitis, rather than a substitute for tooth brushing. In addition, the interval between tooth brushing and applying CHX should be 30 min to 2 h to avoid neutralizing the fluoridation effect of toothpaste [57]. However, even 0.2% CHX is not effective as an adjuvant in carious lesions or in cases of advanced or moderately severe periodontitis [59]. In case of ICU patients, since active oral examinations are required, patients who have difficulty in voluntary communication need a diagnostic tool that can closely assess the oral condition and evaluate it effectively.

#### 3.2.2. Tooth Brushing Method

In 2011, Yao et al., through an RCT, showed that tooth brushing along with the use of distilled water reduced the incidence of VAP and decreased plaque scores compared to that with the use of cotton and gauze. An RCT conducted by the Berry group in 2011 showed that the effect of preventing VAP from additional mouth rinse, such as using sterile water, sodium bicarbonate, and CHX, was not significant if not supplemented by tooth brushing [60]. In another randomized controlled study, it was reported that the effects of dental plaque reduction and VAP prevention in ICU patients undergoing mechanical management of plaque control by tooth brushing were more evident than the effects of mouth rinse, such as Listerine^®^ (Pfizer) or sodium bicarbonate [61]. In 2014, Takeyasu et al. reported that, in patients with oral intubation who received oral hygiene management using an oral moisture gel, the time required for oral hygiene management was shorter and the level of contamination of the intubation cuff was lower [49].

In a 2015 RCT, Ikeda et al. found that, in patients who cannot brush their teeth on their own, electric toothbrushes could help them achieve a significant improvement in plaque score and a significant reduction in oral hygiene management time compared to manual toothbrushes [46]. In 2017, Higashiguchi et al. reported that the use of whole oral wiping and oral nutritional supplements was effective in improving oral hygiene and preventing VAP [48]. Additionally, de Lacerda Vidal et al. reported in an RCT that tooth brushing together with CHX use compared to only CHX use was more effective in reducing the incidence of VAP and the duration of mechanical ventilation [62]. Haghighi et al. reported a remarkable improvement in the plaque index along with effective reduction of VAP occurrence with oral hygiene management while controlling the cuff pressure of the respiratory tract [41]. However, considering that little oral health education is included in the curriculum and that interprofessional education using materials studied by dental hygienists or dentists is poorly performed [63], it is difficult to achieve effective intervention results solely by nurses, and it is necessary to approach this oral management from a dental perspective.

#### 3.2.3. Application of Quantitative Light-Induced Fluorescence Technology

Quantitative light-induced fluorescence (QLF) is a non-invasive in vitro diagnostic tool that uses the principle of autofluorescence reaction of normal hard tissues with blue light in the visible range of 405 nm. QLF can be used to quantify mineral loss and is useful in assessing bacterial lesions, such as caries, plaque, osteomyelitis, and mucosal necrosis, by quantifying red fluorescence derived from bacterial metabolites such as porphyrin [44,64,65,66]. In 2018, Singh et al. evaluated plaque using fluorescence photographs to compare the effects of two oral hygiene management methods [44]. In 2020, Akifusa et al. reported that brushing while visualizing plaque in real time with QLF is very effective in plaque removal [43]. QLF technology can quantify bacterial lesions in the oral cavity to evaluate the oral health and visualize the effects of oral care in real time.

### 3.3. Outcomes of Oral Management of ICU Patients

Many studies have shown the effectiveness of oral hygiene management, including the application of CHX, for respiratory disease prevention in ICU patients [6,11,21,41,47,48,51,53,54,56,60,61,67,68,69,70,71,72,73,74,75,76]. In 2013, Hillier et al. reported that the education of nurses on oral care, including CHX application and other oral care methods, reduced the incidence of VAP [3]. Dale et al. reviewed 84 studies conducted on patients undergoing oral intubation [77]. Previously, oral hygiene management was performed for the convenience of patients; however, recently, it has been emphasized as an infection control practice to prevent VAP [49,77]. In a study conducted in 2014, ICU patients who received oral care and aggressive dental treatment performed by dentists reported a significantly lower incidence of respiratory infections than did the group that received oral care, including the use of CHX, from nurses (8.7% vs 18.1%, respectively) (Adjusted relative risk, 0.44; 95% confidence interval (CI), 0.20–0.96; *p* = 0.04) [69]. In particular, when oral intubation is performed, the risk of VAP occurrence is high because it serves as a passage through which microorganisms in the oral cavity can be transferred into the airways. Therefore, these patients should receive thorough oral hygiene management [54]. In addition, not only CHX gel application, but also brushing should be performed to reduce VAP occurrence during the mechanical ventilation period [62]. However, in a meta-analysis of four RCTs in 2012, brushing did not significantly affect the incidence of VAP [78].

#### 3.3.1. Effect of Oral Hygiene Management by Dental Experts

In 2011, Tada and Miura reported a decrease in the incidence of pneumonia and mortality due to respiratory diseases in a clinical randomized trial on oral hygiene management by dental experts and showed the protective effect of good oral hygiene against respiratory pathogens [79]. In a meta-analysis conducted in 2016, Sjögren et al. reported that the risk of HAI was significantly reduced when dental workers (dentists or dental hygienists) managed oral hygiene (risk ratio; 0.43; 95% CI, 0.25–0.76; *p* = 0.003) [80]. In a 2018 RCT, Bellissimo-Rodrigues et al. showed that active oral hygiene management (tooth brushing, tongue cleaning, scaling, root planning, caries restoration, and extraction) by dentists could effectively prevent respiratory infections [5]. A meta-analysis published in 2020 also reported that the risk of pneumonia, excluding VAP, was significantly reduced (risk ratio random, 0.65; 95% CI, 0.43–0.98; *p* = 0.03) through the involvement of dental practitioners [10].

Notably, four studies on oral hygiene management by dental experts of ICU patients showed that dental treatment, such as periodontal therapy, and oral hygiene management, such as brushing teeth, were highly effective for HAI prevention [5,58,59,60].

#### 3.3.2. Effects of Oral Hygiene Management Other Than the Prevention of Respiratory Infections

In a study of pediatric patients in a neonatal ICU, the patients who received irrigation with distilled water developed significantly fewer oral Gram-negative bacteria than the control group [81]. The tongue is governed by the hypoglossal nerve, which can stimulate the coughing reflex. In a 2016 RCT, Izumi et al. reported that oral hygiene with tongue cleaning increased coughing ability and peak expiratory flow in elderly patients [82].

In contrast, oral hygiene management activities stimulate the cranial nervous system, which may increase the intracranial pressure. Therefore, the position of the patient and the oral hygiene technique (pattern, intensity, interval) should be well controlled during tooth brushing [83]. Oral pathogenic bacteria are potential risk factors for bloodborne and respiratory infections. However, in 2017, Silvestri et al., in a meta-analysis of five studies involving a total of 1655 patients, showed that the application of CHX to critically ill patients did not affect bloodborne infections and associated mortality [84].

Although many studies have been published on the effect of oral management of ICU patients on respiratory infections, there are few studies on the effects on other organ systems or systemic conditions other than respiratory infections. Further studies on this topic with respect to hospital dentistry are necessary.

## 4. Summary

Maintaining good oral hygiene in ICU patients has a significant preventive effect against VAP. Regarding dental interventions by dental experts (dentists and dental hygienists), the effect of oral hygiene management on the prevention of HAI, including VAP, was enhanced.

However, standard oral hygiene management guidelines have not been established in terms of the most effective concentration of gargling agents, types of hygiene management tools, and methods and frequency of oral care. CHX solution has been used as an adjuvant with tooth brushing in ICU patients.

The proportion of ICU patients with poor oral hygiene status, needing active dental treatment for periodontitis or dental caries, is high. Dental interventions must be included in the oral care protocol for ICU patients since oral hygiene management using brushing and mouthwash alone cannot treat oral bacterial diseases, such as periodontitis and dental caries.

Although various oral hygiene management methods have been suggested for ICU patients, mainly led by nurses, few studies have evaluated oral hygiene effects through the effective oral condition evaluation based on objective dental assessments. In addition, the recently developed QLF technology could be used as an objective and effective tool for evaluating oral condition and oral hygiene effectiveness in ICU patients.

Further studies are needed to establish oral care guidelines that can be applied ICUs and to evaluate the effectiveness and validity of these guidelines.

## Figures and Tables

**Figure 1 jcm-10-03681-f001:**
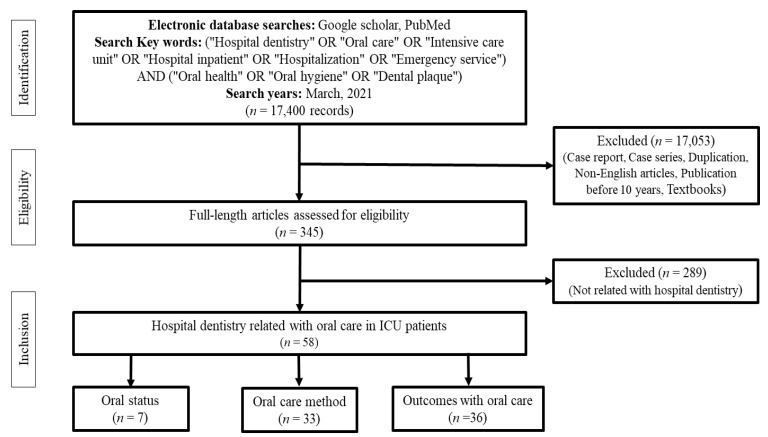
Search strategy.

**Table 1 jcm-10-03681-t001:** Summary of ICU patients’ oral status based on hospital dentistry.

Articles/Types		Subject Information	Index	Average Results
[20]/Review	[27]		Plaque index	At ICU stay day 0: 23% After 10 days: 93%
	[28,29]		Gingivitis	Significantly increased
[21]/Original		Before intervention (*n* = 66)	No. of teeth lost	14.3 ± 8.3
			Probing depth	3.8 ± 1.0 mm
			Plaque index	85.6 ± 20.5%
			Bleeding on probing	48.6 ± 29.7%
[22]/Review	[30]	Patients from England (*n*= 150)	Need for dental operative intervention	75%
			Denture-related candidiasis	38%
			Dental examination within 1 year	15%
	[31]	Patients from Israel (*n* = 225)	Needs of direct dental treatment	65%
			Pseudomembranous candidiasis	56%
	[32]	Patients from New Zealand (*n* = 200)	Need for periodontal intervention	90%
			Need for fillings or extractions	71%
			Carious teeth	1.9
	[33]	Patients from Australia (*n* = 575)	Unhealthy oral condition	76%
			Poor oral hygiene	38%
[5]/Original		Hospitalization within 7 days (*n* = 254)	Gingivitis	54.8%
			Complete edentulism	38.2%
			Periodontitis	29.5%
			Dental caries	29.1%
			Tooth fracture (residual roots)	17.0%
			Mucositis	6.3%
			Oral candidiasis	1.6%
			Odontogenic abscess	0.8%

**Table 2 jcm-10-03681-t002:** Advantages and disadvantages of various oral care methods.

Not Recommended	
Hydrogen peroxide	Irritable, unpleasant taste, and genotoxic
Sodium bicarbonate	Irritation and chemical burns caused by high pH
Topical antibiotics	Changes in the microflora in the oral cavity, unable act against all bacteria that can cause VAP, and risk of developing resistance
Citric acid and glycerin	Can temporarily relieve dry mouth but cause hard tissue demineralization because of low pH
Povidone-iodine	Not effective in reducing plaque and toxic
**Recommended**	
Chlorhexidine	Effective in VAP prevention and plaque control at 0.12–0.2% concentration
Artificial saliva	Beneficial for moisturizing mucosa and maintaining physiological oral flora
Vaseline	Beneficial for moisturizing mucosa and maintaining physiological oral flora
Mechanical plaque control	The most basic and efficient method, a toothbrush is better than a cotton swab, and tooth brushing with toothpaste is more effective

VAP, ventilator-associated pneumonia.

## Data Availability

Not applicable.

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
