# Peer review of "Hospital Dentistry for Intensive Care Unit Patients: A Comprehensive Review"

_jcm, 2021, doi:10.3390/jcm10163681_

Round 1

Reviewer 1 Report

Jun and colleagues provide a review of papers published in the last 10 years on hospital dentistry for ICU patients.  The review tackles an important and underappreciated topic.  There are three foci to the review:  oral health of ICU patients, strategies used to improve oral care in ICU patients, and their impact on outcomes.  The content of the review is good and there’s lots of useful data and citations.  My biggest challenge with the review is that the organization seems haphazard at times.  It jumps back and forth between different topics, approaches, products, and personnel (dental acute intervention, dental health maintenance, different oral care products, impact of toothbrushing, etc.).  I think it would be helpful to organize the review more clearly:  1) oral health status of ICU patients, 2) antiseptics – products and outcomes, 3) toothbrushing – studies and outcomes, and 4) oral care provided by dentists versus nurses – impact on outcomes. 

  1. How did the authors select papers for inclusion? It would be helpful to describe more explicitly in the abstract.
  2. Line 42: what do you mean by “dissemination of pathogens?”
  3. Line 57: what is “non-VAP”
  4. Line 66-67: It’s not surprising that ICU patients are not good at oral hygiene management given their tenuous clinical status, sedation, intubation, etc. Do the authors rather mean that ICU staff are not good at oral hygiene management?  And by good to you mean they lack technical skill or that they do not reliably perform oral hygiene?
  5. Line 72: it’s striking to say that 59% of ICU patients require dental surgery.  Please elaborate.
  6. Can you please clarify what you mean by oral care guidance? Does this refer to guidelines?  Is this a review of strategies that have been proposed and tested?  The term seems nebulous
  7. The authors make no mention of the controversy surrounding oral chlorhexidine and the findings in some studies of ventilated patients that it may be associated with higher mortality rates. This deserves description and analysis.
  8. Similarly, one of the challenges with pneumonia is that the diagnosis is often subjective or non-specific. This has led some authors to advise evaluating the impact of oral care initiatives on objective outcomes (such as duration of mechanical ventilation, ICU length of stay, and mortality) rather than just on pneumonia alone. It would be helpful to consistently describe these outcomes as well.

Author Response

Answers to comments of reviewer #1

Thank you for reviewing this paper. This study reviews the oral hygiene status, oral care guidelines, and outcomes of oral care in intensive care unit (ICU) patients from a dental perspective for effective oral care.

General comments

  • it would be helpful to organize the review more clearly.

⇒ Thank you for your valuable comment. We have corrected the organization of the results section of the manuscript. The order of our results list was changed as follows:

â–¶ The following list of content was changed in ‘Result’ part :

  1. Results

3.1. Assessment of oral status in intensive care unit patients

3.2. Oral hygiene management methods for patients in ICUs

3.2.1. Application of CHX

3.2.2. Tooth brushing method

3.2.3. Application of quantitative light-induced fluorescence technology

3.3. Outcomes of oral management of ICU patients

3.3.1. Effect of oral hygiene management by dental experts

3.3.2. Effects of oral hygiene management other than prevention of respiratory infections

Comments

  1. How did the authors select papers for inclusion? It would be helpful to describe more explicitly in the abstract.

⇒ Thank you for your comment. In the abstract part, the contents of the "Key word selection strategy" for selecting a study suitable for the purpose of this study are mainly described. The details of the selection of article you mentioned are detailed in the Methods section.

â–¶ The following content was described in the methods part [Line 99]:

Three researchers (J.K.K., I.H.K., and S.Y.P.) independently selected the studies based on the following eligibility criteria: type of intervention, type of patients, and type of study. In the first step, titles and abstracts were read for screening of articles. The full text of the selected articles was then analyzed to determine the hospital dentistry's eligibility for the ICU. Any difference of opinions was resolved through discussion with other researchers (M.K.J. and J. K. L.).

  1. Line 42: what do you mean by “dissemination of pathogens?

⇒ In this study, the meaning of “dissemination of pathogens” is as follows: insufficient oral hygiene was shown that oral problems, especially periodontal disease, can act as a focus for dissemination of pathogens with systemic metastatic effect.

  1. Line 57: what is “non-VAP”.

⇒ A pneumonia is classified based on how it is acquired and can be categorized into CAP (community acquired pneumonia), HCAP (healthcare associated pneumonia), HAP (hospital acquired pneumonia), or VAP (ventilator associated pneumonia). Therefore, "non-VAP" in this sentence refers to CAP, HCAP, HAP, which are pneumonias not associated with a ventilator.

We have changed the sentence as below.

â–¶ The following word was added in ‘Introduction’ part [Line 59-60]:

In addition, a systematic review and meta-analysis demonstrated that risk of non-VAP (community acquired pneumonia;CAP, healthcare associated pneumonia;HCAP, hospital acquired pneumonia;HAP) could be reduced by professional dental treatment [10].

  1. Line 66-67: It’s not surprising that ICU patients are not good at oral hygiene management given their tenuous clinical status, sedation, intubation, etc. Do the authors rather mean that ICU staff are not good at oral hygiene management? And by good to you mean they lack technical skill or that they do not reliably perform oral hygiene?

⇒ Thank you for your valuable comment. We have changed the sentence as below.

â–¶ The following sentence was changed in ‘Introduction’ part [Line 66-68]:

However, in previous survey studies targeting ICU nurses, 53–58% of ICU nurses answered that patients had difficulties perform of oral hygiene management due to not receive proper training or materials and instruments are not available, and most of them responded that their oral care was neglected compared to care of other parts of the body [12-15].

  1. Line 72: it’s striking to say that 59% of ICU patients require dental surgery. Please elaborate.

⇒ Thank you for your comment. We have changed the sentence as below.

â–¶ The following word was added in ‘Introduction’ part [Line 77-79]:

In 2016, Amaral et al. reported that 58.7% of ICU patients had one or more need for need invasive dental treatment including periodontal treatment, restorative treatment, surgical treatment, and endodontic treatment [17].

  1. Can you please clarify what you mean by oral care guidance? Does this refer to guidelines? Is this a review of strategies that have been proposed and tested?  The term seems nebulous.

⇒ Thank you for your valuable comment. There was a mistake during the English proofreading process.

We have corrected that word as below.

â–¶ The following word was changed in ‘Abstract’ part [Line 23]:

This study aimed to review the oral hygiene status, oral care guidelines, and outcomes of oral care in intensive care unit (ICU) patients from a dental perspective for effective oral care.

  1. The authors make no mention of the controversy surrounding oral chlorhexidine and the findings in some studies of ventilated patients that it may be associated with higher mortality rates. This deserves description and analysis.

⇒ Most of the articles selected for this review showed that the effect of CHX application had an effect on the reduction of mortality due to VAP. Some studies have shown no difference in mortality reduction. We have added this information in Result section as follows;
â–¶ The following word was added in ‘Result’ part [Line 188-189]:

Some of previous studies reported that the hospital ICU mortality rate associated with VAP was no difference between application of CHX and toothbrush [36,37]. However, a recent review article on RCTs from 2008 to 2018 reported that VAP prevention and oral management of patients are closely related and are important factors in reducing the mortality rate of ICU patients [11].

  1. Similarly, one of the challenges with pneumonia is that the diagnosis is often subjective or non-specific. This has led some authors to advise evaluating the impact of oral care initiatives on objective outcomes (such as duration of mechanical ventilation, ICU length of stay, and mortality) rather than just on pneumonia alone. It would be helpful to consistently describe these outcomes.

⇒ The purpose of this review was to investigate the differences and effects of effective oral care methods for ICU patients from a dental perspective. Through this review, we confirmed that well-performed oral care had a very positive effect on the outcomes (including pneumonia) of ICU patients. However, it was found that many studies did not apply well-controlled oral care and objective oral evaluation indicators.

Therefore, we first suggested the need to improve oral hygiene by preparing oral care guidelines for ICU patients, and to check whether standardized oral care is well performed through objective oral evaluation indicators. we completely agree with your comment that "a prospective study comparing variables of objective outcomes (such as duration of mechanical ventilation, ICU length of stay, and mortality) by providing standardized oral care protocols is needed".

As you mentioned, further studies will also consider whether there are significant differences according to quantitative and objective outcomes such as duration of mechanical ventilation, ICU length of stay, and mortality.

I would like to thank you for your valuable time and considerations, which let our study to have a chance to qualitatively improve.

Reviewer 2 Report

In according to the attached article, we can also take into consideration the correlations between polymicrobial etiology of Odontogenic Abscess and "increased amounts of bacteria obtained from the respiratory tracts of patients with long-term use of mechanical ventilation". 

Author Response

Answers to comments of reviewer #2

Thank you for reviewing this paper. This study reviews the oral hygiene status, oral care guidelines, and outcomes of oral care in intensive care unit (ICU) patients from a dental perspective for effective oral care.

Comments

  1. In according to the attached article, we can also take into consideration the correlations between polymicrobial etiology of Odontogenic Abscess and "increased amounts of bacteria obtained from the respiratory tracts of patients with long-term use of mechanical ventilation"

⇒Thank you for your useful comment. We have added the reference as you recommended.

â–¶ The added reference is below [Line 57]:

[9] Fusconi, M.; Greco, A.; Galli, M.; Polimeni, A.; Yusef, M.; Di Cianni, S.; De Soccio, G.; FR, F.S.; Lombardi, R.; de VINCENTIIS, M. Odontogenic phlegmons and abscesses in relation to the financial situation of Italian families. Minerva stomatologica 2019, 68, 236-241.

I would like to thank you for your valuable time and considerations, which let our study to have a chance to qualitatively improve.

Round 2

Reviewer 1 Report

NA